# Can cognitive training capitalise on near transfer effects? Limited evidence of transfer following online inhibition training in a randomised-controlled trial

**David J. Harris**[1]*, **Mark R. Wilson**[1], **Kieran Chillingsworth**[2], **Gabriella Mitchell**[2], **Sarah Smith**[2], **Tom Arthur**[1], **Kirsty Brock**[1], **Samuel J. Vine**[1]*

1 School of Public Health and Sport Sciences, University of Exeter, Exeter, United Kingdom, 2 Defence Science and Technology Laboratory, Salisbury, United Kingdom

* S.J.Vine@exeter.ac.uk (SJV); D.J.Harris@exeter.ac.uk (DJH)

**Data Availability Statement:** Data cannot be shared publicly because this is a Ministry of

## Abstract

Despite early promise, cognitive training research has failed to deliver consistent real-world benefits and questions have been raised about the experimental rigour of many studies. Several meta-analyses have suggested that there is little to no evidence for transfer of training from computerised tasks to real-world skills. More targeted training approaches that aim to optimise performance on specific tasks have, however, shown more promising effects. In particular, the use of inhibition training for improving shoot/don't-shoot decision-making has returned positive far transfer effects. In the present work, we tested whether an online inhibition training task could generate near and mid-transfer effects in the context of response inhibition tasks. As there has been relatively little testing of retention effects in the literature to date, we also examined whether any benefits would persist over a 1-month interval. In a pre-registered, randomised-controlled trial, participants ($n = 73$) were allocated to either an inhibition training programme (six training sessions of a visual search task with singleton distractor) or a closely matched active control task (that omitted the distractor element). We assessed near transfer to a Flanker task, and mid-transfer to a computerised shoot/don't-shoot task. There was evidence for a near transfer effect, but no evidence for mid-transfer. There was also no evidence that the magnitude of training improvement was related to transfer task performance. This finding adds to the growing body of literature questioning the effectiveness of cognitive training. Given previous positive findings, however, there may still be value in continuing to explore the extent to which cognitive training can capitalise on near or mid-transfer effects for performance optimisation.

## Introduction

Cognitive training has promised much but delivered relatively little in the way of improving human performance. The core principle–that targeted training of domain-general mental

Defence funded project and data are subject to additional restrictions. All relevant code, and the pre-registration document is available online from: https://osf.io/mzxtn/

**Funding:** This work was funded by the Defence Science and Technology Laboratory via the Human Social Science Research Capability framework (HS1.030). The funders contributed to the study design and preparation of the manuscript but had no role in data collection and analysis or decision to publish.

**Competing interests:** The authors have declared that no competing interests exist.

abilities should have benefits for a range of tasks–is appealing for those aiming to optimise human performance [1, 2] or ameliorate deficits arising from clinical disorders, traumatic injury, work-induced fatigue, or age-related decline [3–5]. Yet, findings to date have been mixed, particularly for performance optimisation which was the focus of the present work. Indeed, early promise (e.g., [6]) has given way to increasing questions about the breadth of real-world benefits and the experimental rigour of many studies [7–10]. A series of meta-analyses have suggested that generic cognitive training tasks have benefits for performance on other cognitive tests but return null effects for *far transfer* (i.e., to untrained tasks with demands that only partially overlap with training) [8, 11–13]. Given this weight of evidence against far transfer, the present work explores the idea that there may be value in re-focusing the field of cognitive training towards testing the effectiveness of more targeted and specialised interventions that capitalise on near or mid-transfer effects (i.e., to other cognitive tests that more closely resemble the training task).

Despite some unfavourable findings, there remains substantial interest and ongoing work in the field of cognitive training (to the exasperation of some researchers [14]). One reason for this persistence is that the potential for domain general improvements in cognition remains so alluring. The rigour of much work to date has also been questionable [10, 15]. For example, adequate control groups, pre-registration of analyses, realistic far transfer tests, and assessment of long-term retention have been absent from many studies. This lack of rigour has left open the possibility that better quality work could yet demonstrate the benefits of cognitive training. In a review of methodological standards for cognitive training, Green and colleagues [10] note that the use of a common moniker–'brain training'–for a range of interventions may have also concealed beneficial sub-types. So null effects from meta-analyses could be due to a minority of effective interventions being obscured by an ineffective majority.

While the evidence against true far transfer effects is strong [8, 12, 13, 16, 17], the evidence demonstrating the presence of near transfer effects is also strong [13, 16]. Two key questions in the cognitive training field are, therefore, the degree to which these near transfer effects can be extended (i.e., 'mid-transfer') and whether these effects can provide any practical utility for optimising human performance. In essence, is there any value in designing cognitive training tasks that are closely matched to a target skill to generate very specific performance improvements? There is existing work that has tried to capitalise on what could be termed 'mid-transfer' effects. Rather than aiming to improve domain general intelligence or activities of daily living, this work has focused on training one specific aspect of cognitive function to improve a specific behavioural outcome. The clearest example of this approach is the use of inhibition training to improve the suppression of unsuitable responses during shoot/don't-shoot decision making tasks [18, 19]. Inhibition is a sub-function of working memory which denotes the capacity to obstruct automatic or instinctive responses when they are not appropriate for the context at hand, such as ignoring a distracting noise or delaying a response to threat [20, 21]. Studies which have directly trained inhibition function have generated improved performance in both simulated [22] and live fire shooting tasks [23].

Alternatively, these positive training outcomes may be related to the relationship between inhibition and enhanced attentional capabilities. A recent review by Draheim et al. [24] has argued that attention control ability is more predictive of human performance in real-world tasks than working memory capacity. Draheim et al. describe working memory capacity as the number of units of information an individual can hold in primary memory at once while under cognitive load, while attentional control is the maintenance of goal-relevant behaviour or information and the filtering or blocking of irrelevant and inappropriate information or behaviour. Whilst these two concepts are interrelated, as working memory plays an important role in the control of attention [25], it is possible that researchers seeking real-world

performance improvements should be focusing on attentional control abilities, rather than working memory capacity per se. Contrary to most existing cognitive training interventions which solely target working memory capacity, inhibition training may develop key components of attentional control. It is these potentially adaptive effects that may explain the greater reported success with far transfer tasks [22, 23, 26]. This explanation offers further theoretical support for inhibition training as a promising future route for cognitive training research and indicates that further examination of the underlying mechanisms of inhibition training is needed.

The purpose of this work was to better understand the potential of inhibition training for human performance optimisation. We built upon sports-related findings from Ducrocq et al. [26], where participants were trained on an inhibition task consisting of visual search for tennis ball stimuli accompanied by a singleton distractor that had to be ignored. Ducrocq et al. found inhibition training transferred to i) improved real-world tennis volleying under conditions of performance pressure and ii) better inhibition of visual fixations towards the target, in favour of watching the ball. We adapted this task to test whether computer-based inhibition training (delivered online) could also generate near and mid-transfer effects for shoot/don't-shoot decision making. To date, there has been relatively little testing of retention effects in the literature, so we also sought to test whether any benefits would persist over a 1-month period. Given previous work in this area we hypothesised that individuals in the inhibition training group would outperform those in the active control group on both tests and that these benefits would still be present at a 1-month follow up test.

## Methods

### Pre-registration

The design, hypotheses, and planned analyses for this work were pre-registered prior to any data collection. The pre-registration document is available from the Open Science Framework (https://osf.io/7dv8h). Any analyses that deviate from, or were an addition to, the pre-registered analysis plan are identified as exploratory.

### Design

The study adopted a mixed design, with two independent training groups (full training; active control) completing online cognitive performance assessments at three timepoints (baseline; post-test; retention). Participants were assigned to the following training groups (adapted from [26]):

i.  **Full inhibition training**–computerised visual search task with singleton distractor;

ii. **Active control training**–identical computerised visual search task but with the distracting element omitted.

### Participants

Participants ($n$ = 73, 41 female) were recruited from a student population using opportunity sampling. Favourable opinion was given by the Ministry of Defence Research Ethics Committee (ethical approval was also provided by a University Ethics Committee) before data collection and participants gave written informed consent prior to taking part. The only inclusion criteria were no previous participation in a cognitive training protocol. Based on an a-priori statistical power calculation using G*Power [27], a target sample size of 35 participants per group (70 participants in total) was chosen. The study of Ducrocq et al. [26] reported a large

**Table 1. Summary of group membership (left) and illustration of the flow of participants through the study (right).**

|  | Active Control Group | Training Group |
|---|---|---|
| **Participants** | 36 (15 Male / 21 Female) | 37 (17 Male / 20 Female) |
| **Training adherence**[*] | Incomplete training: $n = 12$ Pre- and post-tests but no retention: $n = 1$ Pre, post, and retention: $n = 23$ | Incomplete training: $n = 12$ Pre- and post-tests but no retention: $n = 4$ Pre, post, and retention: $n = 21$ |

* Participants who completed five out of the six training sessions were included in the final pre-to-post analysis of the near and mid transfer tests, but those with less than five sessions were excluded.

improvement (equivalent to $\eta^2 = .32$) in a mid-level transfer test (anti-saccade task), while other similar work [28] has reported smaller effects ($\eta^2 = .12$) for mid- transfer. A mean of these two effects was used for the power analysis. As a result, to detect an interaction effect of $\eta^2 = .22$ in the main training effect analysis (a 2 (group) x 2 (time) ANOVA), a sample size of 36 (i.e., 18 per group) would be required, given $\alpha = .05$ and power (1-$\beta$) of .85. For the retention test it was estimated that any effects were likely to be smaller in magnitude. We therefore used the smaller effect from Harris et al. [28] as a conservative indicator of a likely effect size. For this part of the study, it was determined that 70 participants (35 per group) would be needed to detect an interaction effect of $\eta^2 = .12$ in a 2 (group) x 2 (time: post v retention) ANOVA, given $\alpha = .05$ and power (1-$\beta$) of .85.

We therefore aimed to recruit 70 participants, and a final sample of 73 participants was achieved (see Table 1). Due to participant attrition, 49 participants out of the initial 73 completed the first phase of the research, but this remained well in excess of our required sample for the main analyses. Only 43 participants completed the retention tests meaning we were only powered to observe larger differences at the retention time-point.

## Tasks and materials

The online cognitive tasks were programmed in PyschoPy [29], an open-source Python-based software platform for experimental psychology studies. The PsychoPy tasks were then uploaded to the online hosting site Pavlovia (https://pavlovia.org/). Participants were sent hyperlinks to the online tasks to access through their web browser. Python code for all the tasks is available from the following GitHub page: https://github.com/Harris-D/Shoot-dont-shoot.

**Near transfer test.** The flanker test is a widely adopted and well validated measure of inhibition ability [30] in which the participant must respond (with a key press) to the direction of a centrally presented arrow. The arrow is flanked by other arrows that are either pointing the same way (congruent) or the opposite way (incongruent) (see Fig 1A, an incongruent trial). Incongruent flanker items require more cognitive effort than congruent items, as they must be ignored by the participant, drawing on the inhibition function of working memory [20]. As this is a pure test of inhibition, it was used to test near transfer effects and whether there is a change in inhibition ability from pre to post inhibition training. The test consisted of 10 practice trials followed by 80 test trials, which were split equally across congruent and incongruent trials, and left and right facing arrows. The left and right arrow keys on the keyboard had to be pressed to indicate the direction of the central arrow. On each trial the arrow stimuli were presented for up to 1500 ms, or until one of the arrow keys was pressed, which would initiate the next trial. The 80-trial block lasted ~5 minutes. Test performance is measured through the size of the congruency effect, that is, the difference in reaction time between presentations with congruent and incongruent distractors (a smaller difference indicates distractions are being inhibited more effectively).

## Illustration of experimental tasks



**Flanker task**

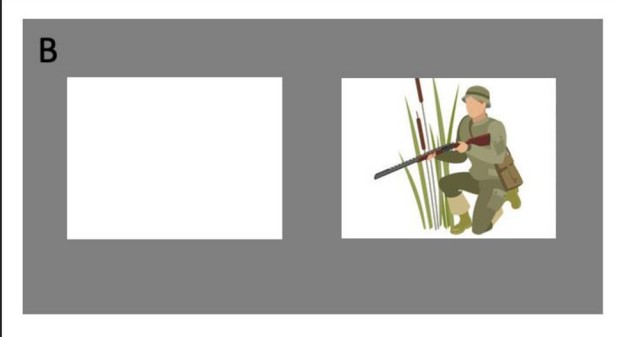

**Shoot/don't-shoot task**

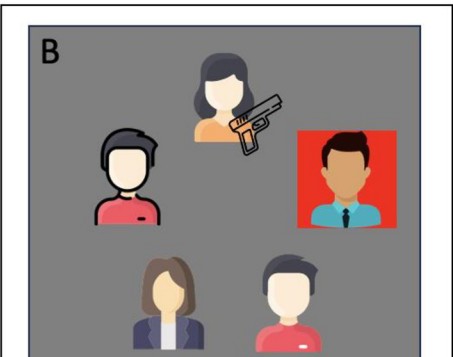

**Training task**

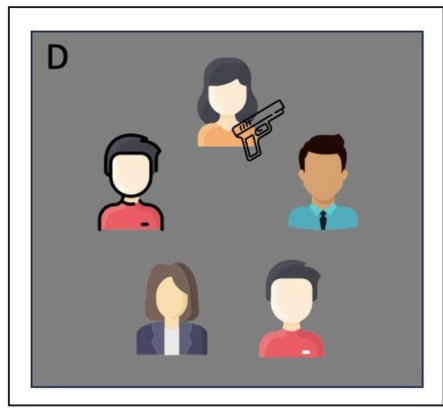

**Active control task**

**Fig 1. Experimental tasks.** Note: A: Flanker task—participant must respond with a key press to indicate the direction of the central arrow. B: Shoot/don't-shoot task—participant must respond with a key press (left/right) to 'shoot' if the person in the image is holding a weapon. The image on the right is just illustrative—real pictures were used for the experimental task. C: Training task—participant indicates whether there is a weapon present in any of the images using a key press. The images shown are illustrative, Sykes-McQueen threat assessment targets were used in the real task. The red distractor slows response times and has to be inhibited. D: Active control task —No distractor is present but the rest of the task is identical to the training task.

### Mid transfer test

A computerised test to further assess inhibition ability was chosen to determine whether the training transferred to a slightly more realistic test of inhibition ability (i.e., mid-transfer). The test was based on a similar shoot/don't-shoot test used in a study by Nieuwenhuys et al. [31], which examined whether Police officers were more likely to shoot when anxious. In the study of Nieuwenhuys et al., participants had to shoot towards one of two locations on a screen if a person with a weapon appeared in that window. The test used here replicates the timings and design of the stimuli used in Nieuwenhuys et al., but instead of shooting a weapon in the direction of the left or right window, participants simply pressed a key to indicate the location of the person with the weapon (if present). The participant was asked to respond as quickly as possible to each image (person with gun present or absent) (see Fig 1B) but were instructed that they should not respond when no weapon was present. On each trial, two empty windows were presented for 1000 ms, then the picture appeared in the left or right window for 500 ms. If a weapon was present but the response was too slow (i.e., >500 ms), the participant received a 'too slow' message. The test consisted of 24 different picture stimuli, 10 of which included a

gun and required a left/right keyboard response using the arrow keys (according to the location on the screen). All stimuli were presented twice each in a randomised order for a total of 48 trials.

In the shoot/don't-shoot task, RTs to weapon present trials and the number of failures to withhold a response when no weapon was present were recorded. To provide greater insight into participants decision-making, the ratio of correct and incorrect decisions was also used to calculate 'd-prime', a measure of detection sensitivity, and 'beta', a measure of response bias. These measures are derived from Signal Detection Theory [32], which outlines methods for describing a person's ability to detect the presence of a signal (which here is the presence or absence of the weapon) amidst the noise (all other aspects of the stimuli and environment that need to be processed). D-prime is calculated from the difference between the z-transformed proportions of hits (H) and false alarms (F): $d' = z(H) - z(F)$, where $H = P(\text{"yes"} \mid \text{YES})$ and $F = P(\text{"yes"} \mid \text{NO})$. It indexes how easily a signal is detected from all the surrounding distractions or complexity (the noise), with higher values indicating more sensitive responding (see Fig 2). Beta is a measure of response bias calculated from the ratio of the normal density

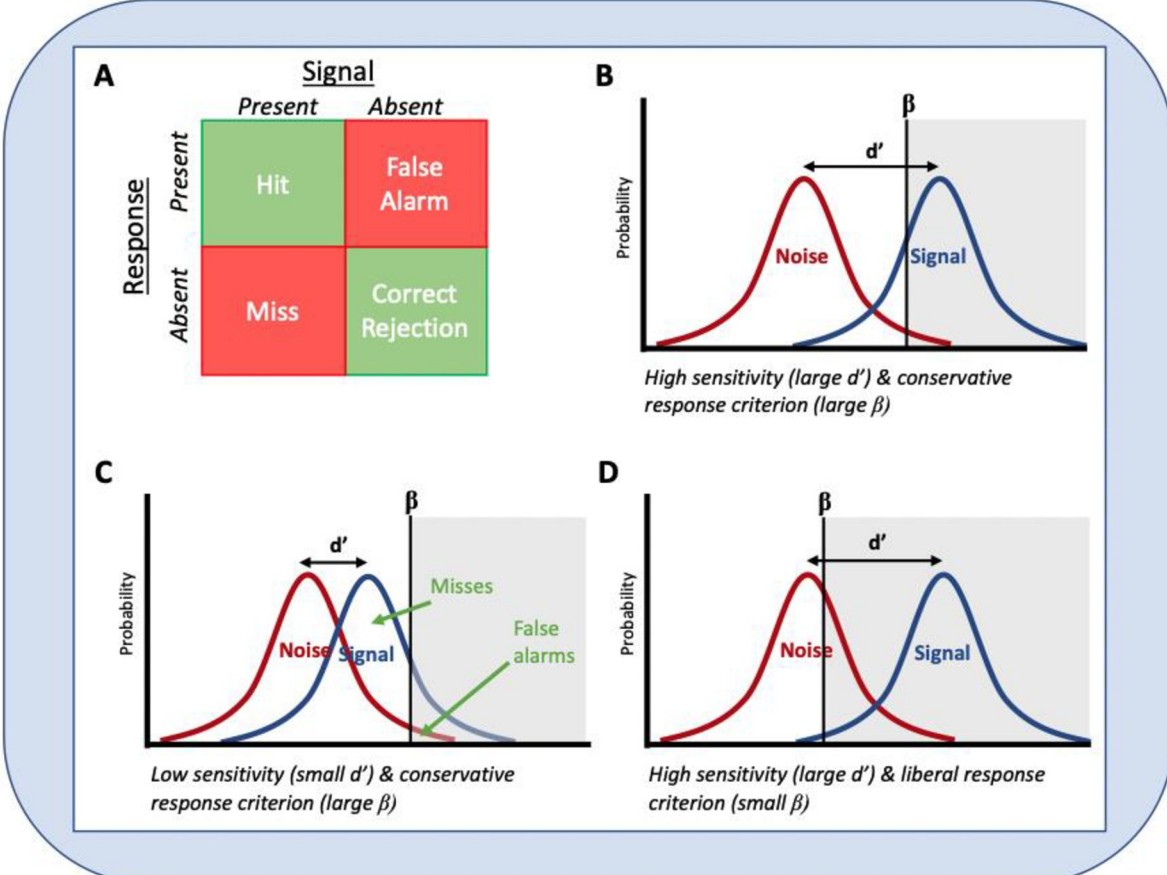

**Fig 2. Illustration of signal detection metrics.** Note: A: The four possible combinations of signals (presence or absence of a threat) and responses (shoot or don't shoot). B-C: D-prime and beta in relation to various distributions of signal (weapon present or absent) and noise. The greyed portion of the figure represents trials where a response was made. In panel B, the participant has a large d-prime value, indicating they were able to perceive a clear difference between the signal and the noise. Beta value is shifted right indicating a conservative response strategy (more 'misses' but very few 'false alarms'). In panel C, d-prime is smaller showing that sensitivity is reduced, but the response strategy is still conservative, so the participant still has few 'false alarms', but a lot more 'misses'. In panel D, d-prime is again large, but this time the response bias is more liberal so there are many more 'false alarms' but few 'misses'.

functions at the criterion of the z-values used in the computation of d-prime. It indicates the relative preference for indicating whether a stimulus is, or is not, present. We employed a standardized response bias criterion, where negative values indicated liberal responding (high hit rates, high false alarms, and few misses) and positive values represented conservative responding (lower hit rates, lower false alarms, but more misses).

## Full training task and active control task

The training and active control tasks were based on inhibition training tasks designed by Ducrocq et al. [26], but adapted for the purposes of shoot/don't-shoot training. In the full training task, the participant was required to indicate if a target image (a person holding a weapon) was present among an array of images as fast as possible, while ignoring a salient distractor item (see Fig 1C). On each trial, a central fixation cross was shown and the image array was presented for 5000 ms, or until a response was given. Each array consisted of 5 images of people holding various items. On 50% of trials one of the five images included a weapon. These images were sourced from the Sykes-McQueen threat assessment 800 series targets (with permission), which are commonly used as threat assessment stimuli in defence and security settings (https://www.mcqueentargets.com/products/#threat). The images in Fig 1 are not the real stimuli and are just illustrative (the real images can be seen on the GitHub page: https://github.com/Harris-D/Shoot-dont-shoot). The weapon present/absent trials also included a singleton distractor on 50% of occasions and were presented in a fully randomised order. The spatial location of the different stimuli, as well as the singleton present/absent and weapon present/absent trials were fully counterbalanced across each individual block. A preceding phase of pilot testing ($n = 7$) was conducted to validate the task by demonstrating that RTs were slowed on singleton present versus singleton absent trials, and that this RT difference reduced over training blocks.

In the active control version of the task, the colour singleton distractor is omitted, providing a perfectly matched visual search task, but without the inhibition demands (see Fig 1D). Consequently, the active control version served to isolate the effect of inhibition practice and accounted for placebo effects arising from participants believing they were assigned to the training group. This made it a very stringent test of the training intervention and answered calls for better matched active control tasks in cognitive training research [10].

Each training session consisted of 80 trials on this task and lasted ~30 minutes. Participants completed six training sessions, on either the full training or active control task, within a 12-day period. The performance metrics for this task were percentage correct responses and RT. The Signal Detection decision making metrics d-prime and beta were calculated for this task in exactly the same way as for the shoot/don't-shoot task.

## Procedure

Potential participants were sent a recruitment email containing a summary of the requirements of the study, a full participant information sheet, and a consent form. Participants were asked to sign and return the consent form via email if they wished to take part in the research. Participants were then randomly assigned to one of the two training groups and sent further instructions on how and when to complete the cognitive tests. Participants were assigned a unique identification number and were instructed to complete the cognitive tasks at consistent times each day, where possible. Participants were asked to complete the six training sessions over a period of 6–12 days, and then repeat the baseline tests (see Fig 3). Next, participants were sent a reminder email to complete the retention test 4 weeks after completing their final training session. Participants were compensated £20.00 for their time.

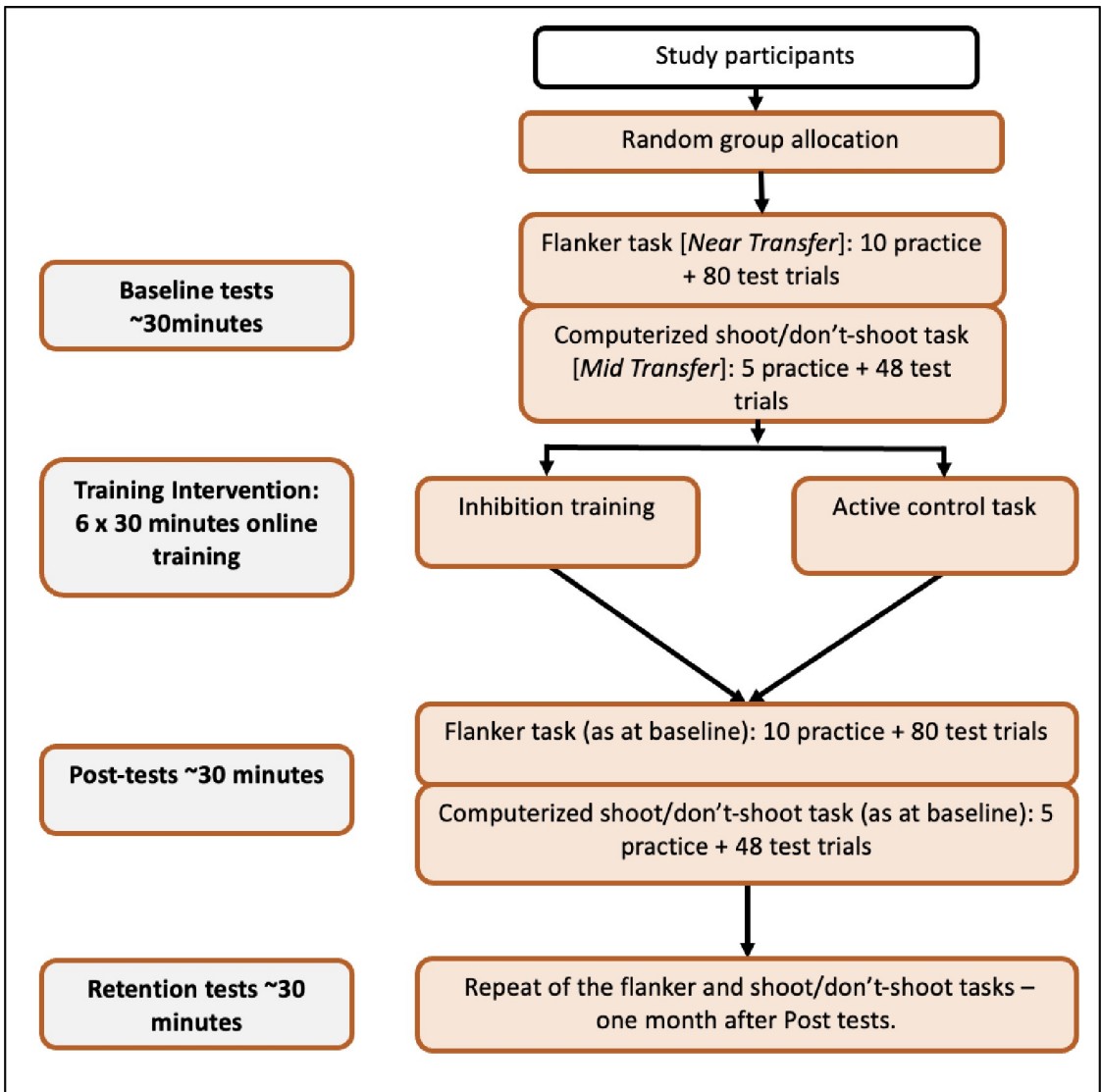

**Fig 3. Trial design.** The figure shows a schematic representation of the flow of participants through the trial.

## Data analysis

Data from the cognitive training tasks was processed using bespoke analysis scripts in MATLAB (2019a; Mathworks, US) which can be found online (https://osf.io/mzxtn/). The derived performance variables were then analysed using JASP (v0.15). Data was screened for outliers and extreme deviations from normality. Outlying values were Windsorised by replacing them with a value 1% larger (or smaller) than the next most extreme value. Some of the performance data were skewed, but as ANOVA is typically robust to such deviations [33], a parametric approach was still used as a method of comparing differences between groups. A series of 2 (time: pre/post) x 2 (group: training/active control) repeated measures ANOVAs were used to examine training effects, and then separate 2 (time: post/retention) x 2 (group: training/active control) repeated measures ANOVAs were used to test for retention. This analysis was performed separately because: 1) it was regarded as a distinct research question, and 2) it enabled participants that did not return for the retention tests to still be included in the

main analysis. An alpha level to determine statistical significance was set at 0.05. The effect size partial eta ($\eta_p{}^2$) was calculated for main effects and Cohen's *d* for t-tests. To support better interpretation of any null effects and supplement the frequentist analysis, we also calculated Bayes Factors, which indicate the relative likelihood of the alternative model compared to the null. We interpret $BF_{10} > 3$ as moderate evidence for the alternative model, and $BF_{10} > 10$ as strong evidence, while $BF_{10} < 0.33$ as moderate evidence for the null and $BF_{10} < 0.1$ as strong evidence for the null [34].

## Results

### Pre to post changes

**Training task.** As a manipulation check, to ensure all participants improved on the training task, regardless of training group, a series of 2 (time: pre/post) x 2 (group: training/active control) ANOVAs were performed to compare performance across the first and last training blocks. Participants significantly improved from pre- to post-training for percentage correct responses, RT, and d-prime. Beta did not significantly change, although it was close to the significance threshold (*p* = .07). There were no group or interaction effects, confirming that there was a parallel training effect for the full inhibition training and active control training tasks (see summary in Table 2 and Fig 4).

As an alternative analysis approach, we re-ran the main analyses using ANCOVAs, testing for group differences at post-training using the baseline scores as a covariate. As some tests showed baseline differences it was decided that this alternative approach could help to determine where reliable training effects were present. These are reported in the supplementary files (https://osf.io/mzxtn/) but largely supported the pre-registered analyses in indicating no benefit of the training.

**Flanker (near transfer).** To assess whether there was a near transfer effect, a 2 (time: pre/ post) x 2 (group: training/active control) ANOVA was performed on the RT difference score from the flanker test (see Fig 5). There was a large effect of training [$F(1,45) = 23.37$, $p < .001$, $\eta_p{}^2 = .34$, $BF_{10} = 1388.82$], indicating a reduction in the RT difference (i.e., better inhibition

**Table 2. ANOVA results for manipulation check.**

| | | *F* | *p* | $\eta_p{}^2$ | $BF_{10}$ |
|---|---|---|---|---|---|
| **Percentage correct** | | | | | |
| | *Time* | 33.10 | < .001 | .41 | $7.05*10^4$ |
| | *Group* | 2.50 | .12 | .05 | 0.61 |
| | *Interaction* | 1.15 | .23 | .02 | 0.45 |
| **Reaction time** | | | | | |
| | *Time* | 120.49 | < .001 | .72 | $1.54*10^{12}$ |
| | *Group* | 3.98 | .052 | .08 | 1.03 |
| | *Interaction* | 0.48 | .49 | .01 | 0.53 |
| **D-prime** | | | | | |
| | *Time* | 32.47 | < .001 | .40 | $5.30*10^4$ |
| | *Group* | 0.02 | .90 | .00 | 0.71 |
| | *Interaction* | 2.79 | .10 | .06 | 0.27 |
| **Beta** | | | | | |
| | *Time* | 3.71 | .07 | .07 | 1.40 |
| | *Group* | 0.04 | .84 | .00 | 0.26 |
| | *Interaction* | 0.00 | .95 | .00 | 0.28 |

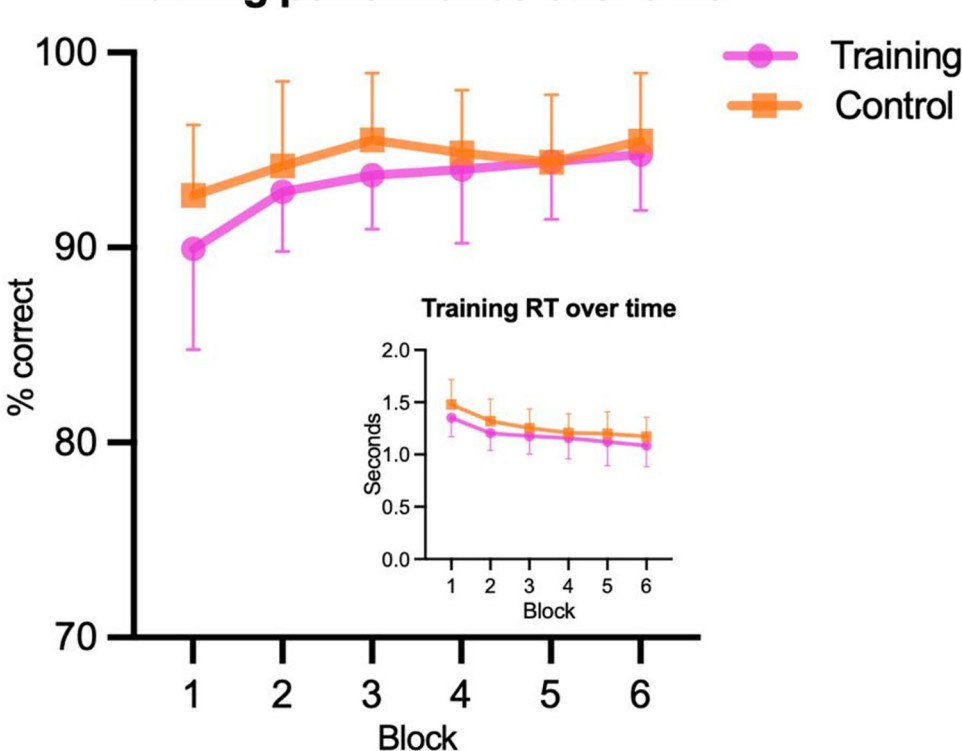

**Fig 4. Training performance.** Plot showing improvement of training performance over time (means and SDs) with corresponding reduction in reaction times (inset).

performance) over time. There was no overall group effect [$F(1,45) = 1.57$, $p = .22$, $\eta_p^2 = .03$, $BF_{10} = 0.57$], but there was a significant group*time interaction [$F(1,45) = 5.10$, $p = .029$, $\eta_p^2 = .10$, $BF_{10} = 2.32$]. Follow-up t-tests with the Bonferroni-Holm correction showed that the interaction effect was driven by a significant reduction in the RT difference for the training group [$t(24) = 5.76$, $p = .002$, $d = 1.16$] but not the control group [$t(21) = 1.58$, $p = .13$, $d = 0.34$]. Further Bonferroni-Holm corrected t-tests at each timepoint showed that there was, however, no difference between the two groups at post-training [$t(45) = 0.17$, $p = .13$, $d = 0.05$, $BF_{10} = 0.31$], or at baseline [$t(45) = 1.89$, $p = .13$, $d = 0.49$, $BF_{10} = 1.15$]. The medium effect for the baseline difference does, however, indicate that the training group started with poorer performance levels (see also *Training Gains Analysis* below).

**Shoot/don't-shoot (mid-transfer).** A series of 2 x 2 ANOVAs from the shoot-don't-shoot task did not suggest any benefit to the training group over the active control group for any of the performance variables (see Fig 6).

**Correct hits.** A 2 x 2 ANOVA on the percentage of targets correctly hit did not indicate any effect of the training, as there was no main effect of time [$F(1,47) = 2.34$, $p = .13$, $\eta_p^2 = .05$, $BF_{10} = 0.59$] or group [$F(1,47) = 0.10$, $p = .76$, $\eta_p^2 = .002$, $BF_{10} = 0.37$], and no interaction [$F(1,47) = 1.40$, $p = .24$, $\eta_p^2 = .03$, $BF_{10} = 0.52$].

**False alarms.** A 2 x 2 ANOVA on the number of non-threat targets hit indicated a small yet significant main effect of time [$F(1,47) = 4.51$, $p = .039$, $\eta_p^2 = .09$, $BF_{10} = 1.22$], but no effect of group [$F(1,47) = 1.26$, $p = .27$, $\eta_p^2 = .03$, $BF_{10} = 0.56$], and no interaction [$F(1,47) = 3.28$, $p = .08$, $\eta_p^2 = .07$, $BF_{10} = 1.04$].

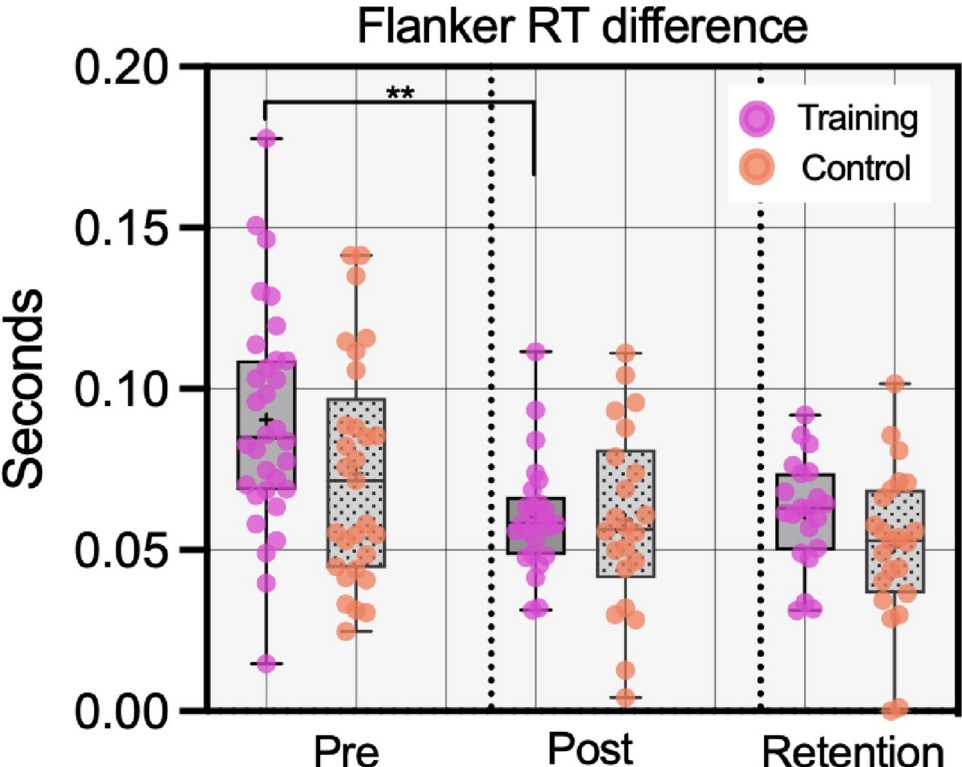

**Fig 5. Box and whisker plots with overlaid data points for performance on the flanker test.** Note: A smaller RT difference shows that the distractors were having a reduced effect, i.e., better inhibition. **p < .01.

**Reaction times.** A 2 x 2 ANOVA on RTs showed no main effect of time [$F(1,46) = 1.74$, $p = .19$, $\eta_p^2 = .04$, $BF_{10} = 0.45$], no effect of group [$F(1,46) = 2.97$, $p = .09$, $\eta_p^2 = .06$, $BF_{10} = 1.05$], and no interaction [$F(1,46) = 0.05$, $p = .82$, $\eta_p^2 = .001$, $BF_{10} = 0.29$].

**D-prime.** A 2 x 2 ANOVA on d-prime coefficients showed a significant main effect of time [$F(1,47) = 7.30$, $p = .01$, $\eta_p^2 = .13$, $BF_{10} = 4.64$], but no effect of group [$F(1,47) = 0.31$, $p = .58$, $\eta_p^2 = .007$, $BF_{10} = 0.38$], and no interaction [$F(1,47) = 0.16$, $p = .69$, $\eta_p^2 = .003$, $BF_{10} = 0.29$].

**Beta.** A 2 x 2 ANOVA on beta coefficients showed no significant main effect of time [$F(1,47) = 0.42$, $p = .52$, $\eta_p^2 = .009$, $BF_{10} = 0.23$], no effect of group [$F(1,47) = 0.69$, $p = .41$, $\eta_p^2 = .01$, $BF_{10} = 0.46$], and no interaction [$F(1,47) = 1.84$, $p = .18$, $\eta_p^2 = .04$, $BF_{10} = 1.14$].

## Retention

Retention of training effects was examined using separate 2 (time: pre/retention) x 2 (group: training/active control) ANOVAs. These analyses are a deviation from the pre-registration which incorrectly outlined comparisons between retention and post-test, instead of retention to baseline. The comparisons of retention and post-test are presented in the supplementary files (https://osf.io/mzxtn/). We also ran analyses using an ANCOVA based approach to test for group differences at retention, using the baseline scores as a covariate. These are reported in the supplementary files (https://osf.io/mzxtn/) but showed no effects of group.

**Flanker task (near transfer).** The 2 x 2 ANOVA showed a main effect of time [$F(1,42) = 17.46$, $p < .001$, $\eta_p^2 = .29$, $BF_{10} = 1700.25$] indicating a large improvement in flanker

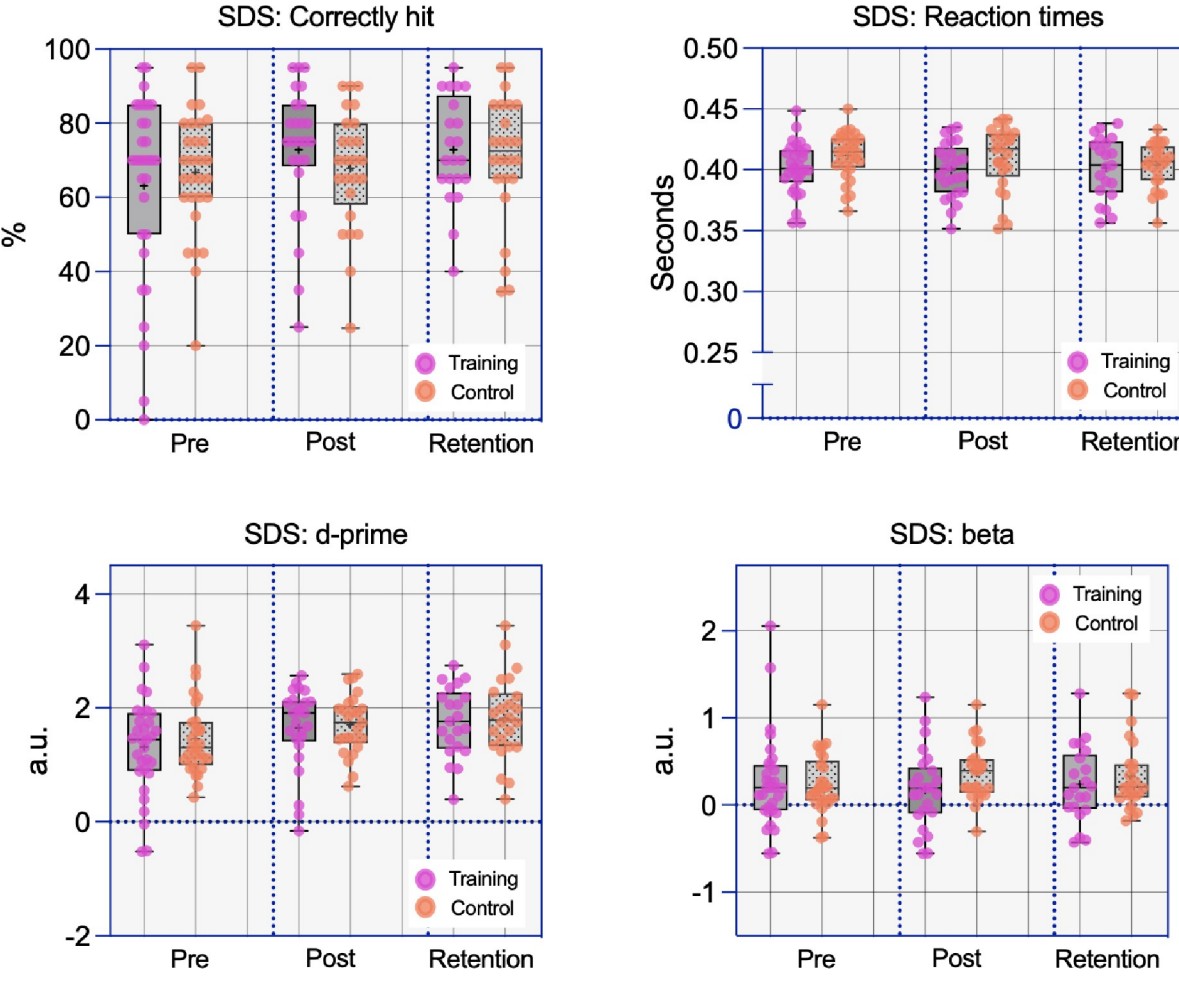

**Fig 6. Box and whisker plots with overlaid data points for performance on the shoot/don't-shoot (SDS) test.**

performance from baseline to retention. There was an effect of group [$F(1,42) = 5.62$, $p = .02$, $\eta_p^2 = .12$, $BF_{10} = 1.09$] reflecting slightly better performance in the control group, but no interaction effect [$F(1,42) = 0.32$, $p = .57$, $\eta_p^2 = .01$, $BF_{10} = 0.34$]. This suggests that large improvements from baseline in both groups were retained over time (see Fig 5), but that there was no benefit of being in the training group.

**Shoot/don't-shoot (mid-transfer).**   Analyses on the shoot/don't-shoot performance variables also indicated general improvements from baseline to retention, but no between-group differences or interactions. This further confirms that there was no benefit of the inhibition training for performance on this task, even after a 1-month interval.

**Correctly hit.**   A 2 x 2 ANOVA indicated an effect of time [$F(1,42) = 8.09$, $p = .007$, $\eta_p^2 = .16$, $BF_{10} = 5.05$], but no effect of group [$F(1,42) = 0.12$, $p = .74$, $\eta_p^2 = .003$, $BF_{10} = 0.40$], and no interaction [$F(1,42) = 1.30$, $p = .26$, $\eta_p^2 = .03$, $BF_{10} = 0.49$].

**False alarms.**   A 2 x 2 ANOVA indicated an effect of time [$F(1,42) = 9.45$, $p = .004$, $\eta_p^2 = .18$, $BF_{10} = 11.21$], but no effect of group [$F(1,42) = 0.28$, $p = .60$, $\eta_p^2 = .007$, $BF_{10} = 0.42$], and no interaction [$F(1,42) = 0.13$, $p = .73$, $\eta_p^2 = .003$, $BF_{10} = 0.31$].

**Reaction times.**   A 2 x 2 ANOVA indicated an effect of time [$F(1,41) = 5.26$, $p = .03$, $\eta_p^2 = .11$, $BF_{10} = 2.19$], but no effect of group [$F(1,41) = 0.61$, $p = .44$, $\eta_p^2 = .02$, $BF_{10} = 0.52$], and no interaction [$F(1,41) = 3.31$, $p = .08$, $\eta_p^2 = .08$, $BF_{10} = 0.90$].

**Table 3. Correlation coefficients (Spearman's Rho) for the relationships between 'training gain' and improvement on the transfer tests.**

| | Training group | | | Control group | | |
|---|---|---|---|---|---|---|
| | Training improvement | | | | | |
| | % Correct | RT | d-prime | % Correct | RT | d-prime |
| **Flanker** | | | | | | |
| *RT difference* | $r = .05, p = 1.00$ | $r = .00, p = 1.00$ | $r = -.05, p = 1.00$ | $r = .30, p = 1.00$ | $r = .44, p = .60$ | $r = .31, p = 1.00$ |
| **Shoot/don't-shoot** | | | | | | |
| *% Correct* | $r = .25, p = 1.00$ | $r = .15, p = 1.00$ | $r = .43, p = .56$ | $r = -.45, p = .56$ | $r = -.22, p = 1.00$ | $r = .01, p = 1.00$ |
| *False alarms* | $r = .01, p = 1.00$ | $r = .01, p = 1.00$ | $r = .06, p = 1.00$ | $r = -.10, p = 1.00$ | $r = -.01, p = 1.00$ | $r = -.06, p = 1.00$ |
| *Response time* | $r = -.02, p = 1.00$ | $r = .17, p = 1.00$ | $r = -.08, p = 1.00$ | $r = -.39, p = .88$ | $r = .01, p = 1.00$ | $r = -.18, p = 1.00$ |
| *d-prime* | $r = -.09, p = 1.00$ | $r = .28, p = 1.00$ | $r = .13, p = 1.00$ | $r = -.36, p = 1.00$ | $r = -.23, p = 1.00$ | $r = .08, p = 1.00$ |

**D-prime.** A 2 x 2 ANOVA indicated an effect of time [$F(1,42) = 17.14$, $p < .001$, $\eta_p^2 = .29$, $BF_{10} = 147.35$], but no effect of group [$F(1,42) = 1.09$, $p = .30$, $\eta_p^2 = .03$, $BF_{10} = 0.48$], and no interaction [$F(1,42) = 0.51$, $p = .48$, $\eta_p^2 = .01$, $BF_{10} = 0.35$].

**Beta.** A 2 x 2 ANOVA indicated no effect of time [$F(1,42) = 0.05$, $p = .82$, $\eta_p^2 = .001$, $BF_{10} = 0.23$], no effect of group [$F(1,42) = 0.00$, $p = .98$, $\eta_p^2 = .000$, $BF_{10} = 0.44$], and no interaction [$F(1,42) = 1.16$, $p = .29$, $\eta_p^2 = .03$, $BF_{10} = 0.49$].

## Training gains analysis

Finally, we conducted an exploratory analysis to examine the relationship between 'training gain' and transfer as it has been suggested that transfer effects might be related to the size of improvement on the training task [35]. We calculated an improvement score for the performance variables from the training task and near and far transfer tasks, based on changes from block 1 to 6 or pre to post (where a positive score corresponded to a relative improvement). This was not done for beta which refers to a response tendency which is not easily characterised as 'better' or 'worse'. The correlation coefficients are summarised in Table 3, but there was little evidence that size of training gain was related to size of improvement on the flanker or shoot/don't-shoot tasks. After a Holm-Bonferroni correction for multiple tests none of the correlations were significant.

## Discussion

In this pre-registered randomised-controlled trial, we examined the effectiveness of an online inhibition training task for improving performance on two tests of inhibition–a flanker task and a shoot/don't-shoot task. We adapted an inhibition training task previously reported in Ducrocq et al. [26] to a military judgemental training context, given the previous success of studies focusing on this skill [22, 23]. Considering the limited assessment of retention of cognitive training effects in existing literature, we sought to examine whether any improvements to inhibition performance persisted over a 1-month period. In short, our results provided little support for the effectiveness of this cognitive training intervention; while there was some evidence for a training benefit on the flanker test (near transfer), there was no evidence of transfer to the shoot/don't-shoot task. This finding adds to the growing body of literature suggesting that cognitive training may support near transfer effects, but that far transfer is unlikely, even when the transfer tests are closely aligned to the training task.

In line with our hypothesis, we found evidence for a near transfer effect. A group by time interaction effect was observed for the flanker task, which was driven by a significant improvement in the training group but not the control group. Inspection of the plots (see Fig 5), as

well as supplementary ANCOVA analyses, indicated that this effect may have been partly due to the training group starting from a poorer baseline. One explanation for the interaction effect is therefore that the cognitive training may have had benefits for those with poorer initial performance and enabled them to 'catch up'. The training gains analysis did not, however, provide support for this explanation. Another possibility is that the interaction was due to small baseline differences and a regression to the mean effect. Consequently, while there was evidence for near transfer this should not be treated as conclusive evidence. Similarly, there was no evidence that participants assigned to the inhibition training group outperformed the control group on any of the shoot/don't-shoot variables. Indeed, Bayes factors often provided weak to moderate support for the null. Assessments of performance at a 1-month follow up also found no difference between the training and control groups, suggesting that benefits did not emerge at a later time.

One reason for the lack of clear training effects could be the very stringent control group employed here. It has been observed that the more positive findings in cognitive training research have tended to originate from studies that have used weaker designs and less well-matched control groups that don't equate input of time and effort or the expectation of a training benefit [9, 10]. However, while a lack of well-matched active control groups is common in the cognitive training field, previous studies finding benefits of inhibition training have tended to use robust controls. Ducrocq et al. [26], for instance, employed the same procedure as in the present work, simply omitting the singleton distractor, and Hamilton et al. [23] used generic working memory tasks as a control comparison to visual search and inhibition tasks. Therefore, the active control task cannot explain why the present results diverge from these previous studies.

Another reason for the limited transfer effects could be that the test conditions were not sufficiently stressful or challenging to reveal the benefits of the training. There is evidence to suggest the benefits of cognitive training may only be revealed when cognitive capacity is already diminished. For instance, Ducrocq et al. [26] found that inhibition training resulted in improved tennis volleying performance, but only when participants were placed under performance pressure. Similarly, Wood et al. [36] found that in a Stroop handgun shooting task (where the colour word determines which target to aim at), low working memory individuals showed significant reductions in shooting accuracy when anxious, while those with high working memory capacity did not. Finally, the transfer tests reported in Hamilton et al. [23] consisted of live fire shooting tasks, which are likely to have posed a much greater stress than the task used here, which was performed in the comfort of people's own homes.

A final consideration that could have impacted the possibility for transfer is that participants in the present study were from a student (i.e., civilian) population. Positive inhibition training effects have been observed in novice or naïve groups [22] but other studies have used tennis players [26] or police officers [23], who could be considered relatively more 'expert'. Future work should therefore consider whether transfer effects are more likely to be revealed in groups who are already well trained in the target skill. In summary, despite the null effects reported here, there may still be value in targeting near or mid-transfer effects for performance optimisation, particularly under more straining or stressful test conditions with an expert participant group.

It is also important to acknowledge the possibility that even though inhibition training has generated more promising findings than traditional working memory capacity training, there are only a small number of studies in this specific research topic. As such, the positive reported effects could be a product of the same publication bias and file drawer effects that blight much of the field (e.g., see [10, 37]). The results of the current study should also be considered in the context of its relative strengths and weaknesses. The study was well-powered and pre-

registered, providing greater rigour than some previous studies. As an online study we could not, however, ensure that participants performed the training and assessment sessions in a quiet distraction-free environment. Variation in their motivation, arousal levels, or environment could also have added noise to the data, reducing any potential training effects. We also experienced a relatively high dropout rate, which can introduce an element of selection bias and reduce the representativeness of the sample. Dropout analyses (reported in the supplementary files: https://osf.io/mzxtn/) suggested that withdrawal from the study was not related to either gender or baseline cognitive ability. To enable more robust conclusions in future work, researchers could implement strategies to improve participant engagement such as providing greater incentives for retention, and carefully considering the study's design to minimize participant burden, making it more likely that participants will complete the trial as intended. Lastly, it is important to note that our findings are more applicable to the use of cognitive training for performance enhancement in healthy individuals than for addressing cognitive deficits in clinical or elderly populations. A limitation is that we did not explicitly screen for neurologic/psychiatric disorders, or collect detailed demographic information, but as the sample consisted of university students this was a predominantly young and healthy sample.

## Conclusions

While research has converged on the idea that there is little evidence for far transfer following practice on computerised 'brain training' tasks, there have been more promising results from methods that specifically focus on the inhibition function of working memory [22, 23, 26]. We have suggested that one reason for these more promising effects could be that these studies are capitalising on 'mid-transfer', where they do not seek domain general improvements in cognition, but performance improvements in one specific task that is closely aligned to the training. While we observed some near transfer, we found no evidence to support the effectiveness of inhibition training for shoot/don't-shoot decision-making. Given previous positive findings there may still, however, be value in continuing to explore the extent which cognitive training can capitalise on near or mid-transfer effects for performance optimisation.

## Author Contributions

**Conceptualization:** David J. Harris, Mark R. Wilson, Kieran Chillingsworth, Gabriella Mitchell, Sarah Smith, Tom Arthur, Samuel J. Vine.

**Data curation:** David J. Harris, Tom Arthur, Kirsty Brock.

**Formal analysis:** David J. Harris.

**Funding acquisition:** David J. Harris, Mark R. Wilson, Samuel J. Vine.

**Investigation:** David J. Harris, Samuel J. Vine.

**Methodology:** David J. Harris, Gabriella Mitchell, Sarah Smith, Tom Arthur, Kirsty Brock, Samuel J. Vine.

**Project administration:** Kieran Chillingsworth, Sarah Smith, Samuel J. Vine.

**Supervision:** Mark R. Wilson, Kieran Chillingsworth, Gabriella Mitchell, Sarah Smith, Samuel J. Vine.

**Visualization:** David J. Harris.

**Writing – original draft:** David J. Harris.

**Writing – review & editing:** Mark R. Wilson, Kieran Chillingsworth, Gabriella Mitchell, Sarah Smith, Tom Arthur, Kirsty Brock, Samuel J. Vine.

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
