## [Decision Letter · Decision Letter 0]

29 Jun 2023

PONE-D-23-00368Can cognitive training capitalise on near transfer effects? Limited evidence of transfer following online inhibition training in a randomised-controlled trialPLOS ONE

Dear Dr. Harris,

Thank you for submitting your manuscript to PLOS ONE. After careful consideration, we feel that it has merit but does not fully meet PLOS ONE’s publication criteria as it currently stands. Therefore, we invite you to submit a revised version of the manuscript that addresses the points raised during the review process.

We look forward to receiving your revised manuscript.

Kind regards,

Celia Andreu-Sánchez

Academic Editor

PLOS ONE

“This work was funded by the Defence Science and Technology Laboratory via the Human Social Science Research Capability framework (HS1.030).”

3. We note that Figure 1 in your submission contain copyrighted images. All PLOS content is published under the Creative Commons Attribution License (CC BY 4.0), which means that the manuscript, images, and Supporting Information files will be freely available online, and any third party is permitted to access, download, copy, distribute, and use these materials in any way, even commercially, with proper attribution. For more information, see our copyright guidelines: http://journals.plos.org/plosone/s/licenses-and-copyright.

b.If you are unable to obtain permission from the original copyright holder to publish these figures under the CC BY 4.0 license or if the copyright holder’s requirements are incompatible with the CC BY 4.0 license, please either i) remove the figure or ii) supply a replacement figure that complies with the CC BY 4.0 license. Please check copyright information on all replacement figures and update the figure caption with source information. If applicable, please specify in the figure caption text when a figure is similar but not identical to the original image and is therefore for illustrative purposes only.

Reviewers' comments:

Reviewer's Responses to Questions

**Comments to the Author**

1. Is the manuscript technically sound, and do the data support the conclusions?

Reviewer #1: Yes

Reviewer #2: Partly

2. Has the statistical analysis been performed appropriately and rigorously? 

Reviewer #1: Yes

Reviewer #2: No

3. Have the authors made all data underlying the findings in their manuscript fully available?

Reviewer #1: Yes

Reviewer #2: Yes

4. Is the manuscript presented in an intelligible fashion and written in standard English?

Reviewer #1: Yes

Reviewer #2: Yes

5. Review Comments to the Author

Reviewer #1: The present study aimed to investigate whether an online inhibition training task could generate near and mid-transfer effects in the context of response inhibition tasks. Furthermore, the authors examine whether any benefits would persist over a 1-month interval. This study was pre-registered and used a randomized controlled trial design. Overall, n=73 participants were included in this study and allocated to either an inhibition training program (six training sessions of a visual search task with a singleton distractor) or a closely matched active control task (that omitted the distractor element). As a result, the authors report tentative evidence for near transfer and no proof of mid-transfer. Furthermore, there was no evidence that the magnitude of training improvement was related to transfer task performance.

The study focuses on a scientific topic that interests readers of PLOS ONE. The English language is decent for being published. However, I am not a native speaker, so this should be checked elsewhere. The introduction has a straightforward structure and is well-written. The design of the study and the methods are described and conducted well and allow replication of the study. Also, the results are reported clearly and transparently. Finally, the discussion also is well-conducted. Accordingly, I only have some minor aspects that could be considered for a potential revision:

- There was a relatively high drop-out rate. Did the authors consider a drop-out analysis to investigate whether any characteristics of the participants could result in the incompletion of the training?

- On page 19, some parts of the manuscript are mixed up at the end of the page.

- The authors did not report any limitations. Therefore, concededly, this study is well-conducted. However, were there any aspects that could be seen as a limitation and could help readers for future studies that should be mentioned?

Reviewer #2: This is a well-written, rigorous study (in terms of design, in particular) of the kind that are much needed in the cognitive training field. I have a few comments that I hope will help improve the manuscript:

1. Given the variability in findings in healthy vs. clinical populations (e.g., far transfer effect results have been more promising in aging/mild cognitive impairment (Basak et al., 2020; Hill et al., 2017)), I think it would be useful from the start to clarify what your target population/target outcome is because it provides important context for your framing. Particularly given the fact that your participants are healthy younger adults; it seems to me that your goal is optimizing performance in healthy younger adults, but this only becomes clear in the last paragraph of the introduction. You should also caveat any interpretations of your finding that they may not generalize to clinical/older adult populations.

2. Related to the point above, it is not clear if this is a healthy sample. Given your very open inclusion/exclusion criteria, did you take any measure of potential neurological/psychiatric disorders? I think that including a population sample is fine, but it may be a limitation in terms of being able to understand the generalizability of your findings.

3. What is the ages of the participants? Again, this is important for interpretation and understanding the generalizability of findings, and is just generally always reported. If age can’t be reported or wasn’t collected, please state that clearly as a limitation.

4. Given that this was a (likely mostly) healthy younger population with a low-risk/non-invasive task, the retention rate seems very poor. Can you explain the poor retention rate, or is this more common for online trials? I think that performing the trial online has some benefits, but the limitations (e.g., retention rate, potentially lower participant engagement) should be clearly outlined. You should also mention that it is online at all opportunities including in the introduction.

5. Given the poor retention during training, how did you deal with missing data? I think it might be appropriate to perform an intention-to-treat analysis or something similar to see if drop-out characteristics affected results.

6. Have you considered performing an analysis such as GEE, MLM (Ma et al., 2012), or an analysis of change score controlling for baseline differences (Mattes & Roheger, 2020)? These may be more robust than an ANOVA, and the GEE and MLM can handle missing data as exists in your study. They also account for baseline differences which would make interpretation easier (may allow you to rule out catch up effects. Also, please confirm that the ANOVA you performed was repeated measures? It isn’t clear in the paper. I also think these more complex models may allow you to model the training gains analysis in a more rigorous way compared to just correlating change scores: I also think that this analysis only needs to be completed for effects that were significant in the main analysis, this might help power given the correction for multiple comparisons.

7. Given you found a group*time interaction on a transfer variable (flanker task) in a pre-registered study with very minimal differences between the conditions in a relatively small sample, your general interpretation is slightly confusing: it seems like you are trying to downplay the result. A group*time interaction is stronger evidence than a post-test t-test, which I would not include. Ideally, the analyses I have suggested will allow for clearer results, but in general I think this is a very promising findings that doesn’t match the tone in which it is described.

References

8. Basak, C., Qin, S., & O'Connell, M. A. (2020). Differential effects of cognitive training modules in healthy aging and mild cognitive impairment: A comprehensive meta-analysis of randomized controlled trials. Psychology and aging, 35(2), 220.

9. Hill, N. T., Mowszowski, L., Naismith, S. L., Chadwick, V. L., Valenzuela, M., & Lampit, A. (2017). Computerized cognitive training in older adults with mild cognitive impairment or dementia: a systematic review and meta-analysis. American Journal of Psychiatry, 174(4), 329-340.

10. Ma, Y., Mazumdar, M., & Memtsoudis, S. G. (2012). Beyond repeated-measures analysis of variance: advanced statistical methods for the analysis of longitudinal data in anesthesia research. Regional Anesthesia & Pain Medicine, 37(1), 99-105.

11. Mattes, A., & Roheger, M. (2020). Nothing wrong about change: the adequate choice of the dependent variable and design in prediction of cognitive training success. BMC Medical Research Methodology, 20(1), 1-15.

6. PLOS authors have the option to publish the peer review history of their article (what does this mean?). If published, this will include your full peer review and any attached files.

Reviewer #1: **Yes: **PD Dr. Jan Christopher Cwik

Reviewer #2: No

---

## [Author Response · Author response to Decision Letter 0]

24 Jul 2023

We wanted to thank the two expert reviewers for taking the time to appraise our work and provide helpful comments on the manuscript. We have made changes to the paper and provide point-by-point responses to each comment below. We think the changes have improved the manuscript and hopefully address any of the reviewers’ concerns. 

Response: Manuscript reformatted. 

“This work was funded by the Defence Science and Technology Laboratory via the Human Social Science Research Capability framework (HS1.030).”

Response: Amended, thanks. 

3. We note that Figure 1 in your submission contain copyrighted images. All PLOS content is published under the Creative Commons Attribution License (CC BY 4.0), which means that the manuscript, images, and Supporting Information files will be freely available online, and any third party is permitted to access, download, copy, distribute, and use these materials in any way, even commercially, with proper attribution. For more information, see our copyright guidelines: http://journals.plos.org/plosone/s/licenses-and-copyright.

Response: We have removed the copyrighted images and replaced them with icons to avoid any reproduction issues. 

Response: Checked, thanks. 

Reviewers comments

Reviewer #1: The present study aimed to investigate whether an online inhibition training task could generate near and mid-transfer effects in the context of response inhibition tasks. Furthermore, the authors examine whether any benefits would persist over a 1-month interval. This study was pre-registered and used a randomized controlled trial design. Overall, n=73 participants were included in this study and allocated to either an inhibition training program (six training sessions of a visual search task with a singleton distractor) or a closely matched active control task (that omitted the distractor element). As a result, the authors report tentative evidence for near transfer and no proof of mid-transfer. Furthermore, there was no evidence that the magnitude of training improvement was related to transfer task performance.

The study focuses on a scientific topic that interests readers of PLOS ONE. The English language is decent for being published. However, I am not a native speaker, so this should be checked elsewhere. The introduction has a straightforward structure and is well-written. The design of the study and the methods are described and conducted well and allow replication of the study. Also, the results are reported clearly and transparently. Finally, the discussion also is well-conducted. Accordingly, I only have some minor aspects that could be considered for a potential revision:

Response: Thank you for the positive comments about the work. 

- There was a relatively high drop-out rate. Did the authors consider a drop-out analysis to investigate whether any characteristics of the participants could result in the incompletion of the training?

Response: Thanks, we hadn’t considered this and it would be an interesting analysis to add. However, we don’t really have that much information on participant characteristics on which to base this analysis. We collected minimal demographic information (only gender) which was probably an error. We will definitely record more data on this in the future to enable this sort of analysis. 

- On page 19, some parts of the manuscript are mixed up at the end of the page.

Response: The formatting might have gone awry when it was changed to a PDF, but it seems to just be the continuation of the footnote from page 18. It looks fine on the resubmitted version.

- The authors did not report any limitations. Therefore, concededly, this study is well-conducted. However, were there any aspects that could be seen as a limitation and could help readers for future studies that should be mentioned?

Response: Thanks, yes of course any study has limitations. We have added some to the discussion as suggested (lines 409-418):

“The results of the current study should also be considered in the context of its relative strengths and weaknesses. The study was well-powered and pre-registered, providing greater rigour than some previous studies. As an online study we could not, however, ensure that participants performed the training and assessment sessions in a quiet distraction-free environment. Variation in their motivation, arousal levels, or environment could also have added noise to the data, reducing any potential training effects. Lastly, it is important to note that our findings are more applicable to the use of cognitive training for performance enhancement in healthy individuals than for addressing cognitive deficits in clinical or elderly populations. A limitation is that we did not explicitly screen for neurologic/psychiatric disorders, or collect detailed demographic information, but as the sample consisted of university students this was a predominantly young and healthy sample.”

Reviewer #2: This is a well-written, rigorous study (in terms of design, in particular) of the kind that are much needed in the cognitive training field. I have a few comments that I hope will help improve the manuscript:

1. Given the variability in findings in healthy vs. clinical populations (e.g., far transfer effect results have been more promising in aging/mild cognitive impairment (Basak et al., 2020; Hill et al., 2017)), I think it would be useful from the start to clarify what your target population/target outcome is because it provides important context for your framing. Particularly given the fact that your participants are healthy younger adults; it seems to me that your goal is optimizing performance in healthy younger adults, but this only becomes clear in the last paragraph of the introduction. You should also caveat any interpretations of your finding that they may not generalize to clinical/older adult populations.

Response: Thanks, we have added an earlier mention in the introduction that our focus here is on performance optimisation rather than addressing cognitive impairment (see line 7) as well as noting that the results don’t apply to clinical or older populations (line 414-416). 

2. Related to the point above, it is not clear if this is a healthy sample. Given your very open inclusion/exclusion criteria, did you take any measure of potential neurological/psychiatric disorders? I think that including a population sample is fine, but it may be a limitation in terms of being able to understand the generalizability of your findings.

Response: We did not ask for this information, but yes this would have been useful to check. Given that the population was university students we have a pretty good idea that this was a broadly young and healthy population. We have added a note on this (line 416-418). 

3. What is the ages of the participants? Again, this is important for interpretation and understanding the generalizability of findings, and is just generally always reported. If age can’t be reported or wasn’t collected, please state that clearly as a limitation.

Response: Added, thanks. 

4. Given that this was a (likely mostly) healthy younger population with a low-risk/non-invasive task, the retention rate seems very poor. Can you explain the poor retention rate, or is this more common for online trials? I think that performing the trial online has some benefits, but the limitations (e.g., retention rate, potentially lower participant engagement) should be clearly outlined. You should also mention that it is online at all opportunities including in the introduction.

Response: It is hard to pinpoint the reason for the drop out. Six sessions is quite a lot and the online nature means it is just very easy for people to decide that they don’t want to do it any more. Most of the drop out occurred after the first session – if people logged on for the second session they generally made it to the end. We have added some extra mentions of the online data collection as suggested. 

5. Given the poor retention during training, how did you deal with missing data? I think it might be appropriate to perform an intention-to-treat analysis or something similar to see if drop-out characteristics affected results.

Response: Thanks, we thought this was a nice idea and we looked into how to do it. However, the intention to treat analysis only works if the participants returned for the post test. But there was only a single participant that didn’t complete all the training but still did the post-test. Essentially, if people got bored and stopped the training, they didn’t return so there’s no data to look at. 

6. Have you considered performing an analysis such as GEE, MLM (Ma et al., 2012), or an analysis of change score controlling for baseline differences (Mattes & Roheger, 2020)? These may be more robust than an ANOVA, and the GEE and MLM can handle missing data as exists in your study. They also account for baseline differences which would make interpretation easier (may allow you to rule out catch up effects. Also, please confirm that the ANOVA you performed was repeated measures? It isn’t clear in the paper. I also think these more complex models may allow you to model the training gains analysis in a more rigorous way compared to just correlating change scores: I also think that this analysis only needs to be completed for effects that were significant in the main analysis, this might help power given the correction for multiple comparisons.

Response: We did initially consider running ANCOVAs for the main analyses with baseline scores as a covariate. We opted for repeated measures ANOVA in the pre-registration plan so we believe we should to stick to this analysis approach for the main paper. In response to this suggestion (and because there were some baseline differences between the groups) we have also run ANCOVA versions of the main analyses and placed them in a supplementary file. These analyses reinforce that there was no benefit of being in the training group. In fact the only group-level effects that were significant in this analysis indicated that the training group had more ‘false alarm’ responses and a more liberal response bias. 

7. Given you found a group*time interaction on a transfer variable (flanker task) in a pre-registered study with very minimal differences between the conditions in a relatively small sample, your general interpretation is slightly confusing: it seems like you are trying to downplay the result. A group*time interaction is stronger evidence than a post-test t-test, which I would not include. Ideally, the analyses I have suggested will allow for clearer results, but in general I think this is a very promising findings that doesn’t match the tone in which it is described.

Response: We have a slightly different interpretation of this result. Our initial interpretation was that the interaction effect could be due to the baseline differences in the flanker scores and therefore just an artefact. We were therefore wary to talking it up too much. The additional tests that you suggested have been useful in this regard, because when controlling for baseline differences there was no group effect. 

While we take the point that the interaction should not be ignored, there was no difference between the groups at post-training time point. In the context of testing the success of a training intervention this seems to be the stronger indicator – those in the training group were not better off than those in the control group. 

We have re-read the conclusion and have reworded in a couple places because we realise that we probably dismissed the interaction too easily, but we have stuck with our original interpretation that this interaction was probably a result of the baseline differences (see lines 360-377).

---

## [Decision Letter · Decision Letter 1]

4 Sep 2023

PONE-D-23-00368R1Can cognitive training capitalise on near transfer effects? Limited evidence of transfer following online inhibition training in a randomised-controlled trialPLOS ONE

Dear Dr. Harris,

Thank you for submitting your manuscript to PLOS ONE. After careful consideration, we feel that it has merit but does not fully meet PLOS ONE’s publication criteria as it currently stands. Therefore, we invite you to submit a revised version of the manuscript that addresses the points raised during the review process. Please approach the comments made by reviewer 2.

We look forward to receiving your revised manuscript.

Kind regards,

Celia Andreu-Sánchez

Academic Editor

PLOS ONE

Reviewers' comments:

Reviewer's Responses to Questions

**Comments to the Author**

1. If the authors have adequately addressed your comments raised in a previous round of review and you feel that this manuscript is now acceptable for publication, you may indicate that here to bypass the “Comments to the Author” section, enter your conflict of interest statement in the “Confidential to Editor” section, and submit your "Accept" recommendation.

Reviewer #1: All comments have been addressed

Reviewer #2: (No Response)

2. Is the manuscript technically sound, and do the data support the conclusions?

Reviewer #1: Yes

Reviewer #2: Partly

3. Has the statistical analysis been performed appropriately and rigorously? 

Reviewer #1: Yes

Reviewer #2: No

4. Have the authors made all data underlying the findings in their manuscript fully available?

Reviewer #1: Yes

Reviewer #2: Yes

5. Is the manuscript presented in an intelligible fashion and written in standard English?

Reviewer #1: Yes

Reviewer #2: Yes

6. Review Comments to the Author

Reviewer #1: The authors revised the manuscript very carefully and took all of my recommendations into consideration. From my point of view, the paper is of good quality and could be accepted for publication.

Reviewer #2: I have a few responses to the revisions:

Thank you for the changes to the introduction and discussion, I think they improve the paper significantly.

1. I think you should add to the limitations that it had a high drop-out rate and you need to find ways to mitigate that in future work if you want to really understand what is happening.

2. I agree with the first reviewer that at least a drop-out analysis should be performed. Even if you don’t have significant demographic info, you can still see if the participants that dropped out had significant differences on the variables you have measured (pre-test scores, gender, etc.).

3. I agree with the authors that basic intent to treat analyses would not help in this case, but there are versions that work with participants that drop out and do not complete post-test. See this paper for example: https://www.ncbi.nlm.nih.gov/pmc/articles/PMC6022256/. I think it is fine if you don’t want to do this analysis, just wanted to provide some information.

4. I still disagree with the framing of the near transfer finding. Interpreting post-training differences is unhelpful because what you are really interested in is change from pre- to post-test. If you have differences at pre-test then post-test differences are even less useful. Sometimes, post-test t-tests are used because if randomization works correctly you can theoretically say that only post-test differences matter as pre-test differences are controlled by randomization, but as in your case randomization often doesn’t work perfectly (although your baseline differences are non-significant). In your study you have a group*time interaction and significant improvement in the active but not control group. This is a positive finding. I agree with carefully interpreting it given the other analyses, but it’s still evidence for near transfer. This sentence: “Our initial interpretation was that the interaction effect could be due to the baseline differences in the flanker scores and therefore just an artefact” in particular suggests a misunderstanding of the goals of this sort of analysis. If you had baseline differences that were erased at post-test, this could either be regression to the mean (which you mention as an option) or it could be a real intervention effect that is being masked at post-test due to real baseline differences. I recommend against post-test t-tests as meaningful analysis in an intervention design and would remove them. Overall, I don’t think this requires a huge change in the framing (it is a very tentative positive finding given the other results), but I do find it strange that you hypothesized a positive result, got a positive result, but are framing it as a negative result (not a caveated positive result).

5. In the retention analysis, it is common to do pre-test vs. follow-up not post-test vs. follow-up. You want to know if there are differences at follow-up compared to baseline, not if differences emerged after post-test (which is unlikely). I would recommend repeating the retention analysis using pre-test.

7. PLOS authors have the option to publish the peer review history of their article (what does this mean?). If published, this will include your full peer review and any attached files.

Reviewer #1: No

Reviewer #2: **Yes: **Adam Turnbull

---

## [Author Response · Author response to Decision Letter 1]

12 Oct 2023

Response: We would like to thank the reviewers for taking the time to re-evaluate the manuscript. The additional queries have been really useful and we have made some substantial changes/additions to the paper. 

Reviewer 2

Thank you for the changes to the introduction and discussion, I think they improve the paper significantly.

1. I think you should add to the limitations that it had a high drop-out rate and you need to find ways to mitigate that in future work if you want to really understand what is happening.

Response: Yes, thanks for this suggestion. Added to the discussion (see lines 385-391). 

2. I agree with the first reviewer that at least a drop-out analysis should be performed. Even if you don’t have significant demographic info, you can still see if the participants that dropped out had significant differences on the variables you have measured (pre-test scores, gender, etc.).

Response: Thanks, we have added this analysis in a supplementary file and refer to it in the discussion where we discuss the dropout rate. We compared the flanker performance between those that withdrew and those that stayed in the study. There was no indication that baseline cognitive performance was different between those who dropped out and those who remained in the study. There was also no indication that one gender dropped out disproportionately. 

3. I agree with the authors that basic intent to treat analyses would not help in this case, but there are versions that work with participants that drop out and do not complete post-test. See this paper for example: https://www.ncbi.nlm.nih.gov/pmc/articles/PMC6022256/. I think it is fine if you don’t want to do this analysis, just wanted to provide some information.

Response: Thanks this looks really useful. Hopefully the analysis we have done provides some indication that the withdrawals did not bias the sample and we will consider this method in future trials. 

4. I still disagree with the framing of the near transfer finding. Interpreting post-training differences is unhelpful because what you are really interested in is change from pre- to post-test. If you have differences at pre-test then post-test differences are even less useful. Sometimes, post-test t-tests are used because if randomization works correctly you can theoretically say that only post-test differences matter as pre-test differences are controlled by randomization, but as in your case randomization often doesn’t work perfectly (although your baseline differences are non-significant). In your study you have a group*time interaction and significant improvement in the active but not control group. This is a positive finding. I agree with carefully interpreting it given the other analyses, but it’s still evidence for near transfer. This sentence: “Our initial interpretation was that the interaction effect could be due to the baseline differences in the flanker scores and therefore just an artefact” in particular suggests a misunderstanding of the goals of this sort of analysis. If you had baseline differences that were erased at post-test, this could either be regression to the mean (which you mention as an option) or it could be a real intervention effect that is being masked at post-test due to real baseline differences. I recommend against post-test t-tests as meaningful analysis in an intervention design and would remove them. Overall, I don’t think this requires a huge change in the framing (it is a very tentative positive finding given the other results), but I do find it strange that you hypothesized a positive result, got a positive result, but are framing it as a negative result (not a caveated positive result).

Response: Thanks, you make a good argument that we have been overly conservative! We have reworded some sections in the discussion (lines 330-342; 403) to frame this as a positive result with some caveats, rather than dismiss it as a negative result. We have made a small change to the abstract as well to reflect this. 

5. In the retention analysis, it is common to do pre-test vs. follow-up not post-test vs. follow-up. You want to know if there are differences at follow-up compared to baseline, not if differences emerged after post-test (which is unlikely). I would recommend repeating the retention analysis using pre-test.

Response: Thank you, yes this approach makes sense. We have repeated the retention analyses comparing to baseline instead (lines 283-308). We think the other analyses still have some value so we have added them as an online supplementary file. The new analyses don’t alter any of the conclusions.

---

## [Decision Letter · Decision Letter 2]

18 Oct 2023

Can cognitive training capitalise on near transfer effects? Limited evidence of transfer following online inhibition training in a randomised-controlled trial

PONE-D-23-00368R2

Dear Dr. Harris,

We’re pleased to inform you that your manuscript has been judged scientifically suitable for publication and will be formally accepted for publication once it meets all outstanding technical requirements.

Kind regards,

Celia Andreu-Sánchez

Academic Editor

PLOS ONE

Additional Editor Comments (optional):

Reviewers' comments:

Reviewer's Responses to Questions

**Comments to the Author**

1. If the authors have adequately addressed your comments raised in a previous round of review and you feel that this manuscript is now acceptable for publication, you may indicate that here to bypass the “Comments to the Author” section, enter your conflict of interest statement in the “Confidential to Editor” section, and submit your "Accept" recommendation.

Reviewer #2: All comments have been addressed

2. Is the manuscript technically sound, and do the data support the conclusions?

Reviewer #2: Yes

3. Has the statistical analysis been performed appropriately and rigorously? 

Reviewer #2: Yes

4. Have the authors made all data underlying the findings in their manuscript fully available?

Reviewer #2: Yes

5. Is the manuscript presented in an intelligible fashion and written in standard English?

Reviewer #2: Yes

6. Review Comments to the Author

Reviewer #2: Thank you for addressing my comments. I think that the manuscript is much improved and should be published.

7. PLOS authors have the option to publish the peer review history of their article (what does this mean?). If published, this will include your full peer review and any attached files.

Reviewer #2: **Yes: **Adam Turnbull

---

## [Editor Report · Acceptance letter]

23 Oct 2023

PONE-D-23-00368R2 

Can cognitive training capitalise on near transfer effects? Limited evidence of transfer following online inhibition training in a randomised-controlled trial 

Dear Dr. Harris:

I'm pleased to inform you that your manuscript has been deemed suitable for publication in PLOS ONE. Congratulations! Your manuscript is now with our production department. 

Kind regards, 

on behalf of

Dr. Celia Andreu-Sánchez 

Academic Editor

PLOS ONE